# Slow oxidation of magnetite nanoparticles elucidates the limits of the Verwey transition

Taehun Kim[1,2], Sangwoo Sim [2], Sumin Lim[3], Midori Amano Patino [4], Jaeyoung Hong [5,6], Jisoo Lee[5,6], Taeghwan Hyeon [5,6], Yuichi Shimakawa [4], Soonchil Lee[3], J. Paul Attfield [7] & Je-Geun Park [1,2,8✉]

Magnetite ($Fe_3O_4$) is of fundamental importance for the Verwey transition near $T_V = 125$ K, below which a complex lattice distortion and electron orders occur. The Verwey transition is suppressed by chemical doping effects giving rise to well-documented first and second-order regimes, but the origin of the order change is unclear. Here, we show that slow oxidation of monodisperse $Fe_3O_4$ nanoparticles leads to an intriguing variation of the Verwey transition: an initial drop of $T_V$ to a minimum at 70 K after 75 days and a followed recovery to 95 K after 160 days. A physical model based on both doping and doping-gradient effects accounts quantitatively for this evolution between inhomogeneous to homogeneous doping regimes. This work demonstrates that slow oxidation of nanoparticles can give exquisite control and separation of homogeneous and inhomogeneous doping effects on the Verwey transition and offers opportunities for similar insights into complex electronic and magnetic phase transitions in other materials.

[1] Center for Quantum Materials, Seoul National University, Seoul 08826, Republic of Korea. [2] Department of Physics and Astronomy, Seoul National University, Seoul 08826, Republic of Korea. [3] Department of Physics, Korea Advanced Institute of Science and Technology, Daejeon 34141, Republic of Korea. [4] Institute for Chemical Research, Kyoto University, Kyoto 611-0011, Japan. [5] Center for Nanoparticle Research, Institute for Basic Science, Seoul 08826, Republic of Korea. [6] School of Chemical and Biological Engineering, Seoul National University, Seoul 08826, Republic of Korea. [7] Center for Science at Extreme Conditions and School of Chemistry, University of Edinburgh, Edinburgh EH9 3JZ, United Kingdom. [8] Institute of Applied Physics, Seoul National University, Seoul 08826, Republic of Korea. ✉email: jgpark10@snu.ac.kr

Oxidation is a fundamental chemical process used to tune many material properties such as changes of correlated electron behavior, as typified by the Verwey transition in magnetite. The signatures of this transition—drastic changes in the crystal structure, magnetization, electric resistivity, thermal conductivity, and other properties near $T_V = 125$ K[1] —have been studied for over eight decades, but many aspects remain mysterious. Among these is the great sensitivity of the transition to oxidation by incorporating tiny amounts of excess oxygen as $Fe_{3(1-\delta)}O_4$ or equivalent cation doping, e.g., with zinc. Oxygen doping of bulk samples suppresses $T_V$ to minimum reported values near 80 K for $\delta = 0.012$[2–6]. A crossover from a sharp first-order transition to a much broader second-order change was observed near $T_V \approx 100$ K for a critical doping $\delta_c = 0.0039$[2,6]. The first- to second-order change was thought to correspond to disruption of the long-range electronic order. However, recent studies have shown that the complex arrangement of charge, orbital, and three Fe-site trimeron[7] states observed by X-ray crystallography below $T_V$ is preserved in the second-order regime and selective doping of one site was proposed to account for the changeover[8].

Control of the tiny nonstoichiometry in bulk $Fe_{3(1-\delta)}O_4$ samples is very challenging, as high temperatures have to be used to give appreciable oxygen diffusion with samples quenched to ambient conditions for characterization[2–6]. A further issue is that any inhomogeneity in the doping may lead to further suppression of $T_V$ as the Verwey transition is also known to be sensitive to non-hydrostatic stresses at constant doping[9]. In general, it is challenging to disentangle the intrinsic doping effect from any contribution due to inhomogeneity when correlated electron systems are chemically doped. However, nanoparticles of magnetite have been developed extensively in recent years for biomedical applications ranging from magnetic resonance imaging contrast agents to thermal therapy, to destroy cancer cells[10]. Chemical control of nanoparticle growth has also enabled fundamental aspects of the Verwey transition, such as the intrinsic domain size dependence, to be explored at the nanoscale[11–13]. As oxygen diffusion into nanoparticles occurs far more rapidly than into bulk material, this has enabled us to explore the effects of room-temperature oxidation of magnetite.

Here we show that slow oxidation of highly stoichiometric and monodisperse $Fe_3O_4$ nanoparticles reveals an intriguing evolution of the Verwey transition. By exposure to several oxygen partial pressures at ambient but controlled temperature of 30 °C, we found an initial drop of $T_V$ down to 70 K after 45–75 days, followed by a recovery up to 95 K after 160 days. We confirmed that the $T_V$ persists up to 1071 days in all experiments. A simple physical model based on Fick's law and basic assumptions about doping and doping-gradient effects can reproduce the intriguing time evolution. This demonstrates that the persistent 95 K value corresponds to the lower limit for homogenously doped magnetite and hence for the first-order regime. In comparison, further suppression down to 70 K results from inhomogeneous strains that characterize the second-order region. We expect that this work will establish an important way to control chemical doping in nanoparticles and can be further applicable to understand related phenomena for other materials having complex phase transitions.

## Results and discussions

High-quality stoichiometric $Fe_3O_4$ nanoparticles were prepared by a method used in previous studies[11]. The average size of nanoparticles used here is $44 \pm 3$ nm. We note that this sample consists of single magnetic domain particles based on comparison of the coercivity to that reported in a previous study[13]. Samples were exposed to controlled oxygen partial pressures $P(O_2)$ from 0.2 to 2 atm, as described

in "Methods", with magnetization, X-ray diffraction (XRD), nuclear magnetic resonance (NMR) spectra, and heat capacity measurements used to monitor changes. The magnetization for freshly prepared samples shows a sharp drop at 120 K with visible hysteresis demonstrating a first-order Verwey transition. Initial oxidation in air ($P(O_2) = 0.2$ atm) leads to broadening and suppression of the Verwey transition. The thermal hysteresis decreases and is suppressed after 13 days when $T_V = 103.7$ K (see Supplementary Fig. 1), indicating a change from first-order to second-order behavior. Further oxidation leads to increased suppression of $T_V$ and broadening of the Verwey transition, quantified as $\Delta T_V$ from the full-width at half-maximum (FWHM) of the peak in the first derivative of the magnetization ($dM/dT$) up to 78 days (Fig. 1a and see also Supplementary Fig. 2a, b). However, further oxidation beyond 78 days led to a recovery of the Verwey transition, with $T_V$ increasing and $\Delta T_V$ decreasing, until a constant value of $T_V = 95$ K was obtained at around 160 days. This persists thereafter for measured times up to 1071 days. The variations of $T_V$ and $\Delta T_V$ are replicated in measurements of other quantities such as the coercive field ($H_c$) (Supplementary Fig. 2h) and the heat capacity, as plotted in Fig. 1b. The heat capacity shows that the sharp transition seen in the fresh sample becomes broadened upon oxidation up to 84 days but then sharpens as $T_V$ increases until 140 days and the Verwey transition is still visible after 1071 days of oxidation with $T_V = 95$ K. The entropy changes through the Verwey transition were estimated and showed a very similar trend to $T_V$ as shown in Supplementary Fig. 3. Similar variations are observed in XRD, through variations in the (440) peak broadening (Supplementary Fig. 4a–c) and in Fe NMR spectra (Supplementary Fig. 5a, b).

Temporal evolutions of these measurements are shown in Fig. 1, and magnetization, XRD, and heat capacity are shown at four different stages of oxidation in Fig. 2, to illustrate the principal changes clearly. The time dependences of $T_V$ and $\Delta T_V$ are summarized in Fig. 3a, showing good agreement between $T_V$ values estimated from magnetization, XRD, NMR, and heat capacity measurements. Transmission electron microscopy images show that nanoparticle size and morphology have not changed after 92 days of oxidation (Supplementary Fig. 6), whereas Mössbauer spectra show that the characteristic magnetite pattern of $Fe^{2+}$ and $Fe^{3+}$ signals is preserved after 102 days (Supplementary Fig. 7). Although magnetization and heat capacity measurements on the 1071 day sample confirm that the Verwey transition is still present with $T_V = 95$ K, the changes in magnetization and entropy at $T_V$ are much smaller than for the 150-day sample, indicating that further oxidation of the nanoparticles leads to a shell of $\gamma\text{-}Fe_2O_3$ (which has no Verwey transition) surrounding a core of $Fe_{3(1-\delta)}O_4$ with $T_V = 95$ K. This takes place over a longer timescale (~100's of days in this experiment) because of the large miscibility gap between maximally oxidized magnetite ($Fe_{2.96}O_4$) and $\gamma\text{-}Fe_2O_3$ ($Fe_{2.67}O_4$).

The oxidation experiment was repeated under other oxygen pressures of $P(O_2) = 1.0$, 1.1, and 2.0 atm, and in all cases, the same time variation was observed, as shown in Fig. 3b. $T_V$ is suppressed to minimum values of 70–80 K in a time $t_{min}$ and then recovers to the persistent value of 95 K at around $2t_{min}$ for all $P(O_2)$. $t_{min}$ decreases with oxygen pressure, from 72 days under $P(O_2) = 0.2$ atm to 47 days at 2 atm. $t_{min}$ characterizes the diffusion kinetics and assuming that rate of oxidation (proportional to $1/t_{min}$) varies with $P(O_2)^{-n}$ gives $n = 0.18$ from the slope of the inset plot to Fig. 3b, close to the value of 0.25 for the oxidation process as discussed in the literature[6]. The $T_V$ and $\Delta T_V$ values from the four experiments show the same variation of time scaled by $P(O_2)^{-n}$ as shown in Supplementary Fig 8.

The discovered variation of the Verwey transition temperature with oxidation time is surprising as a gradual oxidation process would be expected to lead to a monotonic decrease in $T_V$ to a

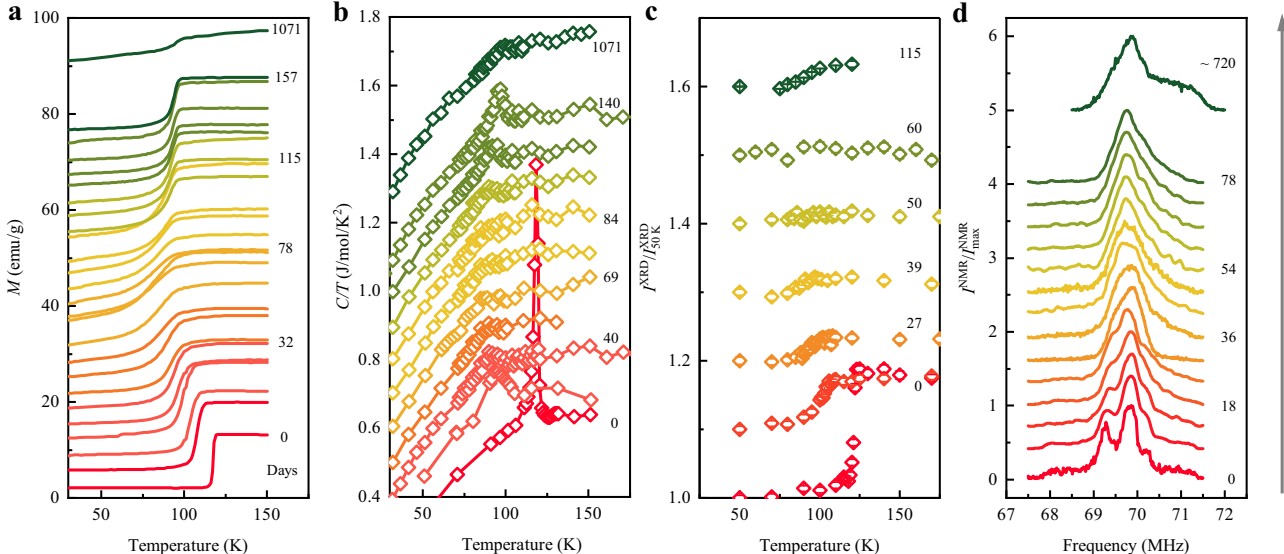

**Fig. 1 Physical measurements for Fe$_3$O$_4$ nanoparticles during up to 1071 days of oxidation. a** Magnetization curves as a function of temperature, with successive data offset by 3.0 emu/g. The magnetization was measured at $H = 100$ Oe. **b** $C/T$ ($C =$ heat capacity) with an offset of 0.1 J/mol/K$^2$. **c** The normalized amplitude of the (440) XRD peak, $I^{XRD}/I_{50K}^{XRD}$, offset by 0.1 units, determined from fits of a Gaussian function to the (440) peak. The error bars are defined as the standard deviation (SD) based on the fitting results of the XRD peak. **d** NMR spectra measured at $T = 80$ K with intensity $I^{NMR}/I_{max}^{NMR}$ normalized to the maximum of each spectrum and offset by 0.3 units. Oxidation times in days are denoted on each plot.

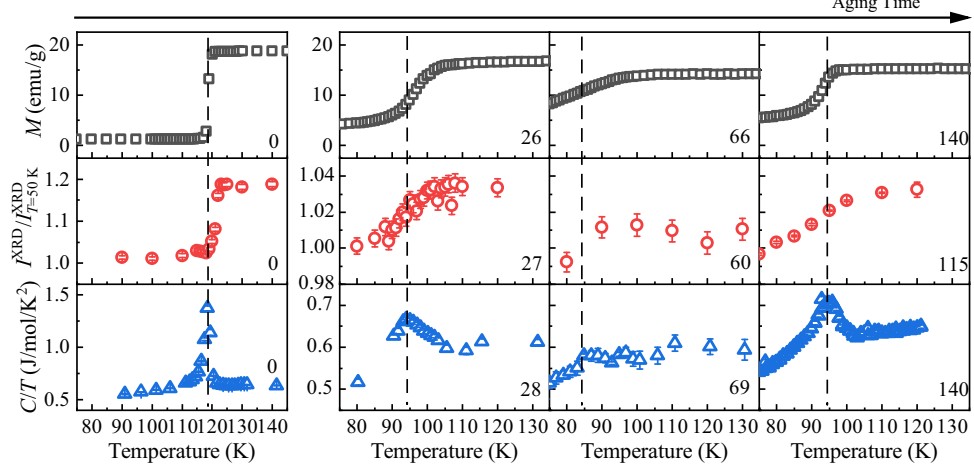

**Fig. 2 Evolution of the Verwey transition in Fe$_3$O$_4$ nanoparticles at four representative stages of oxidation.** The physical properties are as shown in Fig. 1a–c. Data are shown prior to oxidation, after ~27 days when $T_V$ is first suppressed to the 95 K limit for the first-order transition, near $t_{min} = 72$ days where $T_V$ achieves the minimum second-order value and maximum width, and around $2t_{min}$ where a sharper transition is regained with persistent value $T_V = 95$ K. The vertical dashed lines indicate $T_V$ as estimated from the peak fitting in $C/T$ after subtraction of background by fitting. Oxidation times in days are shown on each plot. The error bars for the XRD data are defined as the SD based on the fitting results of the XRD peak. The error bars for the heat capacity data are defined as the SD of the observed heat capacity through the repeated measurements.

minimum value at the maximum doping accommodated by the Fe$_3$O$_4$ lattice. The observation that $T_V$ is initially suppressed by as much as 25 K below the final persistent value of 95 K demonstrates that the transition is suppressed not only by the doping effect from the concentration of added oxygen $C$ but also by inhomogeneous strains created by the oxygen concentration gradient d$C$/d$r$ between the oxygen-rich exterior and oxygen-poor interior of the nanoparticles. This accounts for the maximum $\Delta T_V$ transition width being observed at the minimum $T_V$. We have developed a simple model to quantify these effects using Fick's law of diffusion to simulate d$C$/d$r$, which can readily be transformed to strain (see "Methods"). The calculated total strain $\sigma_{tot}$ variation is well-matched with that of $\Delta T_V$

(Fig. 4a). Furthermore, the overall variation of $T_V$ can be modeled by assuming that $T_V$ is decreased by both a doping term, proportional to $C$, and to a strain term that scales as d$C$/d$r$. Using parameters fitted to the $\Delta T_V$ and $T_V$ values, an excellent fit to the time dependence of $T_V$ is obtained (see Fig. 4b and "Methods"). The fitted diffusion coefficient $D = 2.4 \times 10^{-19}$ cm$^2$/s for these Fe$_3$O$_4$ nanoparticles is consistent with $D \sim 10^{-20}$ cm$^2$/s values reported in the literature[14,15].

The above results demonstrate that ambient temperature exposure of magnetite nanoparticles to oxygen gas provides a simple way to slow down the kinetics of oxidation, enabling the effects of oxygen doping and oxygen-doping gradients on the Verwey transition to be separated. A striking observation is that the persistent value of

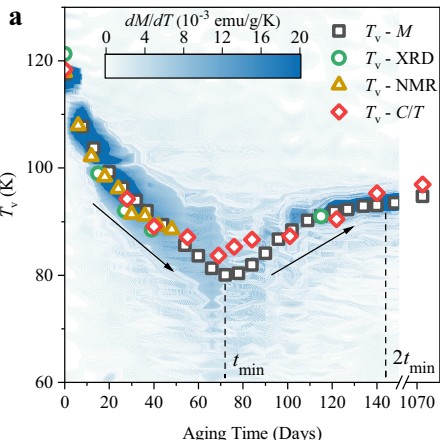
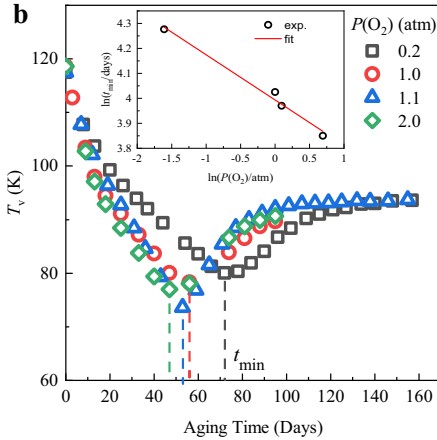

**Fig. 3 Time and oxygen partial pressure dependence of the Verwey transition. a** $T_V$ as a function of oxidation time. The squares, circles, triangles, and diamonds indicate $T_V$ as determined from the magnetization, the XRD measurements, the NMR spectra, and the heat capacity, respectively (details of $T_V$ determination are in "Methods"). The dashed line indicates the oxidation time $t_{min}$, at which the minimum of $T_V$ is observed (in magnetization data). The color-density represents the first derivative of the magnetization, $dM/dT$, which shows how the Verwey transition is broadened around $t_{min}$ but then sharpens as the final $T_V = 95$ K value is reached near $2t_{min}$. (b) Time variations of $T_V$ from magnetization measurements at different oxygen partial pressures. The inset shows the log–log plot of $t_{min}$ against $P(O_2)$ with a linear fit.

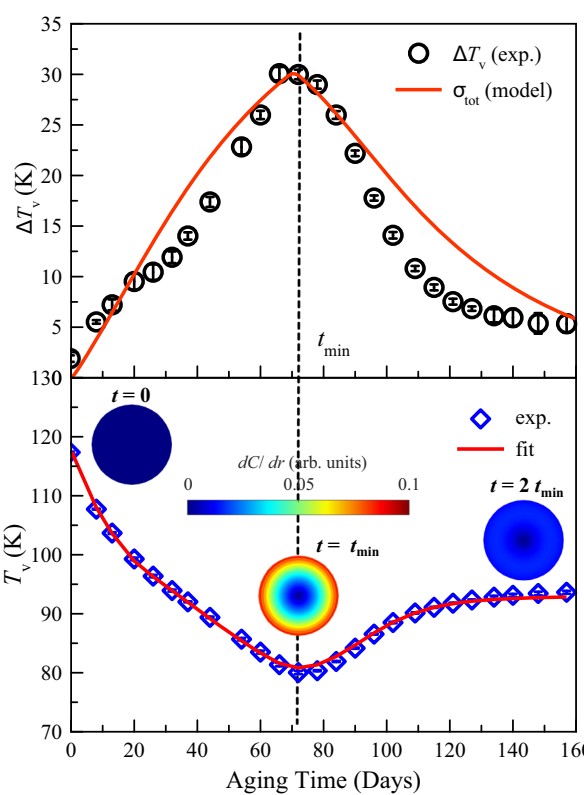

**Fig. 4 Model fits to the Verwey transition temperature and width. a** The aging time dependence of $\Delta T_V$ is defined as the FWHM of the peak in the first derivative of the magnetization ($dM/dT$). The red line represents the calculated total strain $\sigma_{tot}$ (see "Methods"). The absolute value of $\sigma_{tot}$ is scaled to match $\Delta T_V$. The error bars for $\Delta T_V$ is defined as the SD of the fitted FWHM of the peak in $dM/dT$. **b** Time variation of $T_V$ from magnetization points in Fig. 3a, fitted by the model curve as described in "Methods." Drawings show the oxygen concentration gradient $dC/dr$ in magnetite nanoparticles at 0, 70, and 140 days of aging time, calculated for $D = 10^{-19}$ cm²/s, $a = 20$ nm, and $t_{min} = 70$ days using the boundary conditions described in "Methods." The error bars for $T_V$ is defined as the SD of the center position based on the fitting of the peak in $dM/dT$.

$T_V = 95$ K observed at the upper homogenous doping limit for magnetite is not at the lower end of the ~80–120 K range found from previous studies of bulk magnetite and the present nanoparticle results. Instead, it lies close to the $T_V \approx 100$ K crossover between the first- and second-order Verwey transitions. This demonstrates that the crossover corresponds to the intrinsic lower temperature limit of the Verwey transition in homogeneously doped magnetite. Hence, the reported critical doping $\delta_c = 0.0039$ corresponds to the true upper limit for homogenous oxygen doping. Lower $T_V$ values down to 70 K in the second-order regime result from the additional effects of strain gradients on the transition. In the present experiments, these result from the oxygen concentration gradients between the surface and center of the nanoparticles. However, in previous experiments on bulk samples where second-order transitions were reported for higher oxygen-doping levels ($0.0039 < \delta < 0.012$), the incorporation of additional oxygen likely led to the formation of small oxygen-rich regions within the magnetite that gave rise to the strain gradients and nucleate the formation of a secondary $\gamma$-Fe$_2$O$_3$ phase on further oxidation beyond $\delta = 0.012$.

This experimental approach provides a simple way for elucidating doping effects in solid oxides and related materials. Oxygen diffusion into bulk samples at room temperature is prohibitively slow, but the use of nanoparticles where diffusion lengths are reduced to tens of nanometers have enabled the inhomogeneous and homogenous doping of magnetite, and subsequent doping to maghemite, to be separated on timescales of 10's to 100's of days. The rate of oxidation here to homogenous doping level $\delta_c = 0.0039$ in $2t_{min} \approx 140$ days corresponds to the incorporation of only ~70 oxygen atoms per magnetite nanoparticle per day, leading to slow changes in both electronic doping and strain effects. Elastic strain couples to most phase transitions in solids and can lead to large differences in structural phase transition temperatures depending on whether strain is constrained by external stresses ("clamped") or relaxed to thermodynamic equilibrium ("unclamped"). In Landau theory, such changes in strain coupling can drive phase transitions between the second and first order, as observed here for magnetite. Our method of following the dynamics of slow chemical reactions may offer further new insights into how doping and strain effects tune correlated-electron properties in other materials, e.g., transformations of antiferromagnetic LaMnO$_3$ to ferromagnetic

magnetoresistive LaMnO$_{3+\delta}$ and of insulating YBa$_2$Cu$_3$O$_6$ to superconducting YBa$_2$Cu$_3$O$_7$.

In summary, investigation of the Verwey electron-ordering transition during slow oxidation of Fe$_3$O$_4$ nanoparticles up to 1071 days reveals an unusual behavior, with initial suppression and broadening of $T_V$ followed by recovery to a persistent value near 95 K. This variation is explained quantitatively by a simple diffusion model in which both doping concentration and concentration gradient effects are essential. The results account for the previously reported first- and second-order regimes of the Verwey transition as being homogenously and inhomogeneously doped, respectively. Such slow experiments, corresponding to the incorporation of only tens of atoms per nanoparticle per day, are likely to give further insights into other chemical tuning processes of correlated-electron materials.

## Methods

**Sample preparation and oxidation condition.** Stoichiometric and homogenous Fe$_3$O$_4$ nanoparticles were prepared using a standard Schlenk technique[11]. We used thermal decomposition of iron acetylacetonate precursor in the oleic acid surfactant. We varied the precursor-to-surfactant ratios to control the nanoparticles size and we chose 44 nm-diameter nanoparticles with a 3 nm distribution for this work. Samples were stored in an incubator for oxidation by ambient air. The temperature inside the incubator was maintained at around 30 ± 1 °C and the humidity was fixed at ~20%. For the experiments performed at other oxygen pressures, samples were sealed in quartz glass tubes under pressures of $P(O_2) = 1.0$, 1.1, and 2.0 atm.

**Characterizations of the physical properties.** Magnetic susceptibility measurements were conducted using a vibrating sample magnetometer (Lakeshore 7300 series). The samples were cooled to 30 K under zero field and then measured at 100 Oe during heating up to 150 K and cooling down to 30 K in sequence. The isothermal magnetization measurements were also carried out from −6000 to 6000 Oe at 30 and 150 K.

To further support the observations, we examined the structural change induced by the Verwey transition. The XRD experiments were performed using a commercial diffractometer (Bruker D8 Discover System) equipped with an Oxford cryosystem. We mainly focused on the (440) Bragg peak, as it is most drastically modified upon oxidation. The single (440) peak of the cubic phase splits into two peaks when transformed into the monoclinic phase below the Verwey transition. However, it can only be seen as a broad peak for nanoparticle cases, because the XRD peaks are significantly broadened due to size effects (Supplementary Fig. 4a, b). As shown in Fig. 1c, the drop of peak height is no longer observed after 50 days of oxidation but appears again in the data taken after 115 days of oxidation. This is reflected by the FWHM of the (440) peak (Supplementary Fig. 4c).

To examine the oxidation effect on the spin dynamics, we measured the $^{57}$Fe NMR spectra, which are very sensitive to the local field distribution from the Fe ions. NMR spectra of all samples were measured using a home-made solid-state NMR spectroscopy instrument with a cryostat as in the ref. [12]. The single peak located at 69.5 MHz splits into multiple peak structures through the Verwey transition for the fresh sample[12], but the split NMR peaks merge and become a broad peak in the spectra of aged samples as oxidation progresses (Supplementary Fig. 5a, b). The entire oxidation time dependence of the NMR spectra taken at 80 K is summarized in Fig. 1d, where we found that the peak splitting is not observed from 54 days to 78 days. After ~2 years of oxidation, we measured the NMR spectrum and found that it was slightly more structured than that taken on a sample with 78 days of oxidation.

To measure the heat capacity of the Fe$_3$O$_4$ nanoparticles, we used both the Physical Property Measurement System 9 from Quantum Design and a home-built cryostat system. The powdered sample was pelletized before the measurements and mounted onto the sample stage after measuring the addenda. The heat capacity was measured from 30 to 150 K under zero field as shown in Fig. 1b. To define the $T_V$, we first subtracted the background signals using the sum of the polynomial functions. The $T_V$ was defined as the maximum of the remaining heat capacity.

The size distribution and morphology of the samples were characterized by using a JEOL JEM-2010 TEM operating at 200 kV. We could not find any significant difference between the fresh and oxidized samples from the transmission electron microscopy images (Supplementary Fig. 6).

Mössbauer spectra of selected samples were measured below and above $T_V$ in transmission geometry. We used a $^{57}$Co radiation source and the spectra were fitted with Lorentzian functions as described elsewhere[16–18]. Spectra taken after 102 days of oxidation shows a similar structure to that of the fresh sample (Supplementary Fig. 7) and fitted hyperfine fields ($H_{hf}$), quadrupole splitting ($E_Q$), and isomer shift are summarized in Supplementary Tables 1 and 2. These confirm that the oxidized sample is still magnetite and not maghemite or other iron oxides.

**Determination of $T_V$.** We defined $T_V$ from the center of the transition shown in several measurements. In the case of magnetization, the first derivative of the magnetization as a function of temperature ($dM/dT$) was fitted with a Gaussian function giving $T_V$ and $\Delta T_V$ as the center and the FWHM of the peak, respectively. For the XRD, we obtained the (440) peak amplitude as a function of temperature by fitting the data with a Gaussian function and estimated $T_V$ from the center of the peak amplitude drop. $T_V$ was also determined as the mid-point in the curve showing the NMR peak height drop without any fitting. $T_V$ from the heat capacity measurements was taken as the center position of the peak appeared in $C/T$ using Gaussian function. Before the peak fitting, the background of $C/T$ was subtracted based on the background fitting using a polynomial with the formula of $a_0 T^2 + a_1 T^3 + a_2 T^4 + a_3 T^5$.

**Strain $\sigma$ calculation from a diffusion model.** In our theoretical model, the concentration gradient $dC/dr$ is related to oxidation-induced strain. To calculate the total strain $\sigma_{tot}$, we first simulate $dC/dr$ from a simple diffusion model using Fick's law. The diffusion equation in a sphere is[19]:

$$\frac{\partial C}{\partial t} = D\left(\frac{\partial^2 C}{\partial r^2} + \frac{2}{r}\frac{\partial C}{\partial r}\right) \quad (1)$$

To solve the above partial differential equation, we applied the initial and boundary conditions as follows.

$$C = 0, r < a, t = 0 \quad (2)$$

$$C = \begin{cases} t/t_{min} \, at \, r = a \text{ for } t < t_{min} \\ 1 \text{ at } r = a \text{ for } t > t_{min} \end{cases} \quad (3)$$

Here, $C$ is the amount of oxygen diffused into the sphere, with $C = 1$ being fully oxidized. The boundary condition is time dependent and the $C$ at the outer surface is assumed to increase linearly until $t_{min}$ and remains constant after that. By solving the diffusion equation, we obtained $C$ and $dC/dr$ as a function of radius $r$ (Supplementary Fig. 9a). For the calculation results shown in Fig. 4 and Supplementary Fig. 9a, we assumed $a = 20$ nm, $D = 10^{-19}$ cm$^2$/s, and $t_{min} = 70$ days.

As described in the main text, using the simulated $dC/dr$, we calculated the total strain $\sigma_{tot}$ inside nanoparticles. We assume that the $dC/dr$ applies pressure on adjacent oxidized shells. Based on the theory of elasticity[20], the strain $\sigma^i$ applied in the interface between $i$-th and $j$-th shells with hydrostatic pressure $p^i$ is written as below with the assumption that $p^i = p^j$.

$$\sigma^i = -p^i\left(1 + \frac{8r_i^3 r_j^3}{\left(r_i + r_j\right)^3\left(r_j^3 - r_i^3\right)}\right) \quad (4)$$

We assume that the hydrostatic pressure $p^i$ is linearly proportional to $dC/dr$ in the nanoparticle so the total strain $\sigma_{tot}$ can be calculated $\sigma_{tot} = \sum_i \sigma^i$, as shown in Fig. 4a and Supplementary Fig. 9b. Based on the $D$ dependence of $\sigma_{tot}$, the diffusion coefficient was taken to be $D = 10^{-19}$ cm$^2$/s.

**Fitting the oxidation time dependence of $T_V$.** It is known that $T_V$ is linearly dependent on off-stoichiometry parameter $\delta(t)$ in the bulk case[2,17]. Here we assume that in nanoparticles, the additional term $d\delta(t)/dr$ also affects $T_V(t)$, as the strain builds up during the oxidation as described in the main text. As $\delta(t)$ is equivalent to $C(t)$ from the above diffusion model, we write $T_V(t)$ as:

$$T_v(t) = a_1 - a_2 \cdot C(t) - a_3 \cdot \frac{dC(t)}{dr} \quad (5)$$

Here, $a_1 \sim a_3$ are arbitrary constants. As $dC(t)/dr$ scales as strain $\sigma_{tot}$ and transition width $\Delta T_V$ (Fig. 4a), we replace $dC(t)/dr$ with $\Delta T_V(t)$ and $C(t)$ can be assumed to be an exponential decay function, so the equation becomes:

$$T_v(t) = a_1' - a_2'\exp\left(-\frac{t}{\tau}\right) - a_3' \cdot \Delta T_v(t) \quad (6)$$

Using the $\Delta T_V(t)$ values from magnetization measurements, we fit the oxidation time dependence of $T_V(t)$ from Eq. (6) as shown in Fig. 4b. We found that $a_1' = 95.47$ K, $a_2' = 23.01$ K, $a_3' = 0.5$, and $\tau = 19.44$ days. The time exponent should be related to the diffusion coefficient $D$ as $1/\tau = -D\pi^2/a^2$ from the general solution of the diffusion equation in a sphere[19], giving $D = 2.41 \times 10^{-19}$ cm$^2$/s in excellent agreement with the assumed value of $D = 10^{-19}$ cm$^2$/s above.

## Data availability

The data that support the findings of this study are available from the corresponding author J.-G.P. upon reasonable request.

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

## Acknowledgements

We acknowledge helpful discussions with Drs. Jaehong Jeong and Soon Gu Kwon. Work at the Center for Quantum Materials was supported by the Leading Researcher Program of the National Research Foundation of Korea (Grant number 2020R1A3B2079375). This work was partly supported by grants for the International Collaborative Research Program of ICR in Kyoto University and by the JSPS Core-to-Core Program (A) Advanced Research Networks. J.P.A. acknowledges EPSRC for support.

## Author contributions

J.-G.P. initiated the project, while T.K. and J.-G.P. designed the concept for this study with further ideas from J.P.A. J.L. and J.H. synthesized $Fe_3O_4$ nanoparticles under the supervision of T.H. TEM images were taken by J.L. T.K. performed the experimental works on preparing oxidation set-up, measuring magnetization, XRD, and data analysis. Heat capacity was measured by T.K. and S.S. NMR spectra were taken and analyzed by S.M.L. and S.L. Mossbauer data were collected and analyzed by T.K., M.A.P. and Y.S. T.K., J.P.A. and J.-G.P. wrote the manuscript with inputs from all authors.

## Competing interests

The authors declare no competing interests.
