## [Peer Review File · Nature Communications]

Reviewers' Comments:

Reviewer #1:

Remarks to the Author:

In this Article, Kim et al. investigate the suppression of Verwey transition in magnetite by means of slow oxidation of monodisperse Fe₃O₄ nanoparticles studied systematically over an extended period of time. Using a combination of thermodynamic measurements, diffraction, and NMR studies the authors show that the exposure of the nanoparticles to fixed oxygen pressures leads to an initial lowering of the Verwey transition temperature to 70 K followed by a slow recovery to a persistent value near 95 K. Such an unusual variation indicates that the transition is suppressed not only by the oxygen doping but also by inhomogeneous strains created by the oxygen concentration gradient. The authors introduce a simple model based on Fick's law of diffusion that accounts for the observed evolution of the Verwey transition. Finally, they relate the previously reported first- and second-order regimes of the Verwey transition to magnetite being homogeneously and inhomogeneously doped, respectively.

The manuscript is written clearly and concisely. The research appears to be scientifically sound, and the conclusions are well supported by the data. I highly prize the novelty of the authors' approach and I applaud the authors' perseverance in the experiments. The work performed by the authors not only advances our understanding of the effect of oxidation on the Verwey transition in magnetite but also paves the way for similar studies on nanoparticles of other correlated electron systems.

I think the relevance of the presented work is sufficient to publish the manuscript in a highly prestigious journal such as Nature Communications, and I am convinced that the article will attract many citations.

Below are a few minor comments and suggestions for the authors to consider:

- In the Methods section I do not see any mention of the method that was used to investigate specific heat of the nanoparticles.
- In the second paragraph of the Results section the authors mentioned changes in entropy at TV. While I see the C/T vs T plot (Fig. 1b), I do not find any plot or table illustrating directly changes in entropy associated with the Verwey transition. I would welcome such estimates. Perhaps the authors could even use them to evaluate the thickness of a γ -Fe₂O₃ shell?
- I do not see any mention in the text that the studied nanoparticles are in the single domain size range.
- I would welcome a brief comment on the agreement of the observed coercivity and saturation magnetization with the bulk values reported for Fe₃O₄ and γ -Fe₂O₃.
- I would find it helpful to have the information about the magnetic field used for the magnetization study ($H = 100$ Oe) not only in the Methods section but also in captions of Fig. 1 as well as Figs 1 and 2 of the Extended Data.

Reviewer #2:

Remarks to the Author:

As noted, the Verwey transition in magnetite is both the oldest noted magnetic phase transition in history, as well as remaining scientifically enigmatic. The transition is exquisitely sensitive to doping, yet this is not obviously an electronic phenomenon alone. This paper is a crisp analysis of the doping effects which reveals that the couplings may depend indirectly on the elastic strain (and its inhomogeneity). The novelty is that by using the oxidation of nanoparticles the authors are able to study annealing that in bulk would take ridiculously long times.

The results show that a first order transition rapidly broadens and the mean T_c is lowered *below* the final state - which has a much sharper, second-order-like transition. A simple but convincing phenomenological model helps to explain this: oxygen diffusion first yields an inhomogeneous

strain distribution that then at longer times anneals to become more uniform. Somehow this enables the system to reform in a different thermodynamic limit. This is a result that is well digested and clear; it is likely very important not for this system alone but for many electronic and structural phase transitions where slow processes can influence the structural reorganisation; and it is likely to be fodder for materials theorists.

Editorialising somewhat. Elastic strain couples to nearly all electronic phase transitions, and it has long been understood that there can be large differences in T_c in structural phase transitions depending on whether the strain is fixed by geometry and external stresses ("clamped") or reaches thermodynamic equilibrium ("unclamped"). This paper demonstrates that such effects are important even in a classic magnet where the textbook theory is purely electronic. At the level of Landau theory, such strain couplings can drive phase transitions between second and first order for example. The handle of using nanoparticles and dynamics of oxidation may well throw light on many different systems.

Of course it remains to be understood exactly why such tiny changes in doping can have remarkably large effects. This paper shows how entirely 'mechanically' - in the literal sense of the word - these are leveraged.

Reviewer #3:

Remarks to the Author:

The current manuscript by Kim et al reports a very careful study on the slow oxidization of Fe_3O_4 nanoparticles over a period up to 3 years. A variety of probes, including magnetic and specific heat measurements, x-ray diffraction, NMR, TEM, and Mossbauer, were employed to monitor the effect of oxidization on the Verwey transition of Fe_3O_4 nanoparticles. Instead of a monotonic suppression of the transition temperature, the authors observed a V-shaped time dependence. The modelling helps explain this interesting time dependence and distinguish the doping from doping-gradient effects. Once strain field induced by inhomogeneous doping plays a role, the transition turns to be 2nd order. This explains the oxygen-doping dependence of the Verwey transition observed in bulk samples. This beautiful work deserves publication if the authors can further explain how the impact of this work meets the criteria of Nature Communications. The authors mentioned at the end of the main text that this slow oxidization approach may be applied to other systems to resolve how chemical doping and strain field affect the correlated electron properties. Could the authors elaborate more on this?

For those particles with $T_v=95\text{K}$ after an extended duration in oxygen, is the Verwey transition 1st order or 2nd order? The authors mentioned that the particles might have a Fe_2O_3 shell. Will this core-shell structure induce any strain field?

The authors collected x-ray powder diffraction patterns on samples at different stages. Is it possible to get some estimate of the strain by the Williamson-Hall analysis?

Report of the First Referee

General comment:

In this Article, Kim et al. investigate the suppression of Verwey transition in magnetite by means of slow oxidation of monodisperse Fe₃O₄ nanoparticles studied systematically over an extended period of time. Using a combination of thermodynamic measurements, diffraction, and NMR studies the authors show that the exposure of the nanoparticles to fixed oxygen pressures leads to an initial lowering of the Verwey transition temperature to 70 K followed by a slow recovery to a persistent value near 95 K. Such an unusual variation indicates that the transition is suppressed not only by the oxygen doping but also by inhomogeneous strains created by the oxygen concentration gradient. The authors introduce a simple model based on Fick's law of diffusion that accounts for the observed evolution of the Verwey transition. Finally, they relate the previously reported first- and second-order regimes of the Verwey transition to magnetite being homogeneously and inhomogeneously doped, respectively.

The manuscript is written clearly and concisely. The research appears to be scientifically sound, and the conclusions are well supported by the data. I highly prize the novelty of the authors' approach and I applaud the authors' perseverance in the experiments. The work performed by the authors not only advances our understanding of the effect of oxidation on the Verwey transition in magnetite but also paves the way for similar studies on nanoparticles of other correlated electron systems.

I think the relevance of the presented work is sufficient to publish the manuscript in a highly prestigious journal such as Nature Communications, and I am convinced that the article will attract many citations.

Reply:

We thank the referee for his/her appreciation of our work and its broader impact.

Comment #1:

In the Methods section I do not see any mention of the method that was used to investigate specific heat of the nanoparticles.

Reply:

Thank you for the comment. We now added a new description of our method for the heat capacity measurements to the Methods section.

Comment #2:

In the second paragraph of the Results section the authors mentioned changes in entropy at TV. While I see the C/T vs T plot (Fig. 1b), I do not find any plot or table illustrating directly changes in entropy associated with the Verwey transition. I would welcome such estimates. Perhaps the authors could even use them to evaluate the thickness of a γ -Fe₂O₃ shell?

Reply:

We appreciate the referee's suggestion. In the revised manuscript, we traced the entropy changes of the Fe₃O₄ nanoparticles upon oxidation. To estimate the entropy due to the Verwey transition,

we used the method used in our previous work: Jisoo Lee et al., Nano Letters 15, 4337 (2015). After subtracting off the nonmagnetic part, we integrated the magnetic heat capacity to get the total entropy of the Verwey transition (see the final results plotted below).

As can be seen in the figure, the changes in the entropy follow a similar pattern to the Verwey transition temperature. For instance, the entropy of the 140 days aged sample is found to be about a third of the original value. However, we are reluctant to conclude that this value should be taken as evidence of γ - Fe_2O_3 shell growing that much at the expense of the Fe_3O_4 . The simple reason is that, as we argued, the non-stoichiometric $\text{Fe}_{3-\delta}\text{O}_4$ can also show a much suppressed Verwey transition with smaller entropy. Therefore, the thickness of the γ - Fe_2O_3 shell would inevitably be overestimated. We added the entropy plot in Extended data Fig. 3 and added new comments to the main text (in the first paragraph of the Results section).

Comment #3:

I do not see any mention in the text that the studied nanoparticles are in the single domain size range.

Reply:

We found that all our Fe_3O_4 nanoparticles are in a single domain. From the previous study reported in Ref. 12, we used the coercivity value to conclude that the single domain is formed in

our Fe₃O₄ nanoparticles up to 120 nm. We added this description to the first paragraph of the Results section.

Comment #4:

I would welcome a brief comment on the agreement of the observed coercivity and saturation magnetization with the bulk values reported for Fe₃O₄ and γ -Fe₂O₃.

Reply:

Thank you for the comment. The reported bulk values of the saturated magnetization for Fe₃O₄ and γ -Fe₂O₃ are 92 and 74 emu/g, respectively. So, the fresh state of the Fe₃O₄ nanoparticle is similar to the bulk values, and the oxidized state shows a reduced magnetic moment compared to the bulk values. It agrees that the Fe₃O₄ nanoparticle converted to γ -Fe₂O₃ upon the oxidation. The coercivity of both Fe₃O₄ and γ -Fe₂O₃ have relatively similar values in the range of 100 ~ 400 Oe. The observed coercivity of the Fe₃O₄ nanoparticles upon oxidation shows good agreement with the reported bulk values.

Comment #5:

I would find it helpful to have the information about the magnetic field used for the magnetization study (H = 100 Oe) not only in the Methods section but also in captions of Fig. 1 as well as Figs 1 and 2 of the Extended Data.

Reply:

We mentioned the magnetic field value in the caption of Fig. 1 and the Extended Data Fig. 1 & 2.

Report of the Second Referee

General comment:

As noted, the verwey transition in magnetite is both the oldest noted magnetic phase transition in history, as well as remaining scientifically enigmatic. The transition is exquisitely sensitive to doping, yet this is not obviously an electronic phenomenon alone. This paper is a crisp analysis of the doping effects which reveals that the couplings may depend indirectly on the elastic strain (and its inhomogeneity). The novelty is that by using the oxidation of nanoparticles the authors are able to study annealing that in bulk would take ridiculously long times.

The results show that a first order transition rapidly broadens and the mean T_c is lowered *below* the final state - which has a much sharper, second-order-like transition. A simple but convincing phenomenological model helps to explain this: oxygen diffusion first yields an inhomogeneous strain distribution that then at longer times anneals to become more uniform. Somehow this enables the system to reform in a different thermodynamic limit. This is a result that is well digested and clear; it is likely very important not for this system alone but for many electronic and structural phase transitions where slow processes can influence the structural reorganisation; and it is likely to be fodder for materials theorists.

Reply:

We are pleased that the referee appreciated our results and the broader impacts upon other related fields.

Comment #1:

Editorialising somewhat. Elastic strain couples to nearly all electronic phase transitions, and it has long been understood that there can be large differences in T_c in structural phase transitions depending on whether the strain is fixed by geometry and external stresses ("clamped") or reaches thermodynamic equilibrium ("unclamped"). This paper demonstrates that such effects are important even in a classic magnet where the textbook theory is purely electronic. At the level of Landau theory, such strain couplings can drive phase transitions between second and first order for example. The handle of using nanoparticles and dynamics of oxidation may well throw light on many different systems.

Reply:

Thank you for the interesting interpretation, which we like. We added new comments in line with the referee's idea.

Comment #2:

Of course it remains to be understood exactly why such tiny changes in doping can have remarkably large effects. This paper shows how entirely 'mechanically' - in the literal sense of the word - these are leveraged.

Reply:

Thank you for your kind and supportive understanding.

Report of the Third Referee

General comment:

The current manuscript by Kim et al reports a very careful study on the slow oxidization of Fe₃O₄ nanoparticles over a period up to 3 years. A variety of probes, including magnetic and specific heat measurements, x-ray diffraction, NMR, TEM, and Mossbauer, were employed to monitor the effect of oxidization on the Verwey transition of Fe₃O₄ nanoparticles. Instead of a monotonic suppression of the transition temperature, the authors observed a V-shaped time dependence. The modelling helps explain this interesting time dependence and distinguish the doping from doping-gradient effects. Once strain field induced by inhomogeneous doping plays a role, the transition turns to be 2nd order. This explains the oxygen-doping dependence of the Verwey transition observed in bulk samples. This beautiful work deserves publication if the authors can further explain how the impact of this work meets the criteria of Nature Communications.

Reply:

We appreciate the referee's valuable assessments. As the other two referees found, we think that our study paves a novel way of studying correlated electron systems using high-quality nanoparticles. We demonstrated that using a nanoparticles system, we could have the clearest

picture of the ‘slow-motion’ nanoscale oxidation. In turn, we showed that this novel approach is a valuable new tool to study the possible coupling between strain and the complex phase transition.

Comment #1:

The authors mentioned at the end of the main text that this slow oxidization approach may be applied to other systems to resolve how chemical doping and strain field affect the correlated electron properties. Could the authors elaborate more on this?

Reply:

Thank you for the helpful question. The room temperature oxidation of Fe₃O₄ in bulk form is too slow to observe any meaningful way. Instead, we showed that if they are high quality, nanoparticles could be a new platform to study many intricate effects of correlated electrons, including chemical doping and strain, by room temperature oxidation. It allows a handy knob to control the inhomogeneity of the strain easily, as we demonstrated. So, our outlook is further to extend our method to other systems with strong correlations. We added new comments to that effect on the last page of the manuscript.

Comment #2:

For those particles with T_v=95K after an extended duration in oxygen, is the Verwey transition 1st order or 2nd order? The authors mentioned that the particles might have a Fe₂O₃ shell. Will this core-shell structure induce any strain field?

Reply:

All our data indicate that the Verwey transition after 1071 days of oxidation remains to be second order. We think that a strain field induced by the lattice mismatch between Fe₃O₄ core and γ-Fe₂O₃ shell: about $\Delta a/a = 7 \times 10^{-4}$, is rather small as compared to the strain from the inhomogeneity.

Comment #3:

The authors collected x-ray powder diffraction patterns on samples at different stages. Is it possible to get some estimate of the strain by the Williamson-Hall analysis?

Reply:

We thank the referee’s comment. Unfortunately, we did not collect the whole XRD patterns in a broader range of two theta angles. We focused instead on the (440) peak, which showed the most drastic changes through the Verwey transition. Thus, we cannot use the W-H analysis for our data.

Reviewers' Comments:

Reviewer #1:

Remarks to the Author:

I am fully satisfied with the revisions made by the authors. I have no further comments and recommend this paper for publication in Nature Communications.

Reviewer #3:

Remarks to the Author:

All my concerns have been well addressed. I would recommend acceptance of this work.

Report of the First Referee

General comment:

I am fully satisfied with the revisions made by the authors. I have no further comments and recommend this paper for publication in Nature Communications.

Reply:

We thank the referee for his/her acceptance of our work.

Report of the Second Referee

General comment:

-

Report of the Third Referee

General comment:

All my concerns have been well addressed. I would recommend acceptance of this work.

Reply:

We thank the referee for acceptance and helpful comments.

Editorial Report

Reply:

We complied with all the requests of the editorial report.